# Developing a multimedia patient-reported outcomes measure for low literacy patients with a human-centered design approach

Chao Long Azad[1,2], Laura K. Beres[3], Albert W. Wu[4], Allan Fong[5], Aviram M. Giladi[1]*

1 The Curtis National Hand Center, MedStar Union Memorial Hospital, Baltimore, Maryland, United States of America, 2 Department of Plastic and Reconstructive Surgery, Johns Hopkins Medicine, Baltimore, Maryland, United States of America, 3 Department of International Health, Johns Hopkins Bloomberg School of Public Health, Baltimore, Maryland, United States of America, 4 Department of Health Policy and Management, Johns Hopkins Bloomberg School of Public Health, Baltimore, Maryland, United States of America, 5 MedStar Health Research Institute, Hyattsville, Maryland, United States of America

* editor@curtishand.com

This is a Registered Report and may have an associated publication; please check the article page on the journal site for any related articles.

## Abstract

### Introduction

Almost all patient-reported outcomes measures (PROMs) are text-based, which impedes accurate completion by low and limited literacy patients. Few PROMs are designed or validated to be self-administered, either in clinical or research settings, by patients of all literacy levels. We aimed to adapt the Patient Reported Outcomes Measurement Information System Upper Extremity Short Form (PROMIS-UE) to a multimedia version (mPROMIS-UE) that can be self-administered by hand and upper extremity patients of all literacy levels.

### Methods

Our study in which we applied the Multimedia Adaptation Protocol included seven phases completed in a serial, iterative fashion: planning with our community advisory board; direct observation; discovery interviews with patients, caregivers, and clinic staff; ideation; prototyping; member-checking interviews; and feedback. Direct observations were documented in memos that underwent rapid thematic analysis. Interviews were audio-recorded and documented using analytic memos; a rapid, framework-guided thematic analysis with both inductive and deductive themes was performed. Themes were distilled into design challenges to guide ideation and prototyping that involved our multidisciplinary research team. To assess completeness, credibility, and acceptability we completed additional interviews with member-checking of initial findings and consulted our community advisory board.

### Results

We conducted 12 hours of observations. We interviewed 17 adult English-speaking participants (12 patients, 3 caregivers, 2 staff) of mixed literacy. Our interviews revealed two distinct user personas and three distinct literacy personas; we developed the mPROMIS-UE with these personas in mind. Themes from interviews were distilled into four broad design

**Data Availability Statement:** All relevant data are within the manuscript and its Supporting Information files.

**Funding:** CLA and AMG received funding from the American Foundation for Surgery of the Hand (Award 3698) and the Johns Hopkins Physician-Scientist Training Program for this work. Each author certifies that there are no funding or commercial associations (consultancies, stock ownership, equity interest, patent/licensing arrangements, etc.) that might pose a conflict of interest in connection with the submitted article related to the author or any immediate family members. The funders had no role in study design, data collection and analysis, decision to publish, or preparation of the manuscript. https://www.afsh. org/s/ https://www.hopkinsmedicine.org/som/ gme/pstp/.

**Competing interests:** The authors have declared that no competing interests exist.

challenges surrounding literacy, customizability, convenience, and shame. We identified features (audio, animations, icons, avatars, progress indicator, illustrated response scale) that addressed the design challenges. The last 6 interviews included member-checking; participants felt that the themes, design challenges, and corresponding features resonated with them. These features were synthesized into an mPROMIS-UE prototype that underwent rounds of iterative refinement, the last of which was guided by recommendations from our community advisory board.

## Discussion

We successfully adapted the PROMIS-UE to an mPROMIS-UE that addresses the challenges identified by a mixed literacy hand and upper extremity patient cohort. This demonstrates the feasibility of adapting PROMs to multimedia versions. Future research will include back adaptation, usability testing via qualitative evaluation, and psychometric validation of the mPROMIS-UE. A validated mPROMIS-UE will expand clinicians' and investigators' ability to capture patient-reported outcomes in mixed literacy populations.

## Introduction

A growing focus on patient-centered and value-based care in the United States (US) has supported the importance of collecting data from the patient perspective [1–3]. The patient perspective can be captured by patient-reported outcomes (PROs), which are assessments of a health condition made directly by the patient without interpretation or estimation by clinicians or anyone else [4]. PROs, collected via patient-reported outcomes measures (PROMs), can be used to guide clinical care, as part of performance measures for care quality, or as an outcome measure in research. Despite the utility, importance, and growing popularity of PROs, essentially all existing PROMs are reliant on text. This represents a substantial methodological and implementation challenge among low and limited literacy patient populations. Up to nearly one-half of US adults have low English literacy [5, 6], and low literacy rates are disproportionately higher in ethnic and racial minority populations [7]. Interviewer-administration is an option, but it is labor intensive and expensive, and introduces potential for interviewer bias [8] and patient embarrassment [9] that are especially important considerations in low literacy populations [10, 11]. It is critical that PROMs are designed for implementation in an inclusive, equitable fashion. Otherwise, reliance on PROs to understand outcomes and care quality risks becoming a systemic means of perpetuating and exacerbating health disparities.

To capture PROs more equitably, we need a self-administered PROM that accommodates a range of literacy levels. This may also allow us to capture PROs more accurately by ensuring that low literacy patients' responses are accurate reflections of their experiences [12]. Because multimedia PROMs (mPROMs) with audiovisual components are a promising solution, we developed the Multimedia Adaptation Protocol (MAP) [13], which adapts validated, text-based PROMs to a multimedia format that supports self-administration in populations with mixed literacy. The MAP includes four stages: forward adaptation, back adaptation, qualitative evaluation, and validation.

To execute and demonstrate the application of the MAP, we elected to adapt the Patient Reported Outcomes Measurement Information System (PROMIS®) Upper Extremity (PROMIS-UE) to a multimedia version (mPROMIS-UE) in a hand and upper extremity surgery

patient population. In the U.S., an estimated 3.5 million patients sustain hand and upper extremity trauma each year [14], with more than 1.1 million upper extremity ambulatory procedures performed annually [15]. In hand and upper extremity care, growing evidence of the divergence between patient and provider perception of health and functional status has driven an increase in the use of PROs as outcome measures in research and to guide patient-centered and value-based clinical care [16–19]. The PROMIS-UE and the hand and upper extremity population are ideal for the first application of the MAP because variation in physical function, such as in upper extremity function, may be easier to convey in multimedia elements than variations in mental or social health.

The goal of this study was to demonstrate the feasibility of adapting PROMs to multimedia versions by executing the first stage of the MAP, forward adaptation, to adapt the PROMIS-UE to the mPROMIS-UE. This would result in the first multimedia PROM that could be self-administered in a mixed literacy patient population.

## Methods

### Research ethics

This study was performed in accordance with the ethical standards in the 1964 Declaration of Helsinki. This study was carried out in accordance with relevant regulations of the US Health Insurance Portability and Accountability Act (HIPAA). This study was approved by the MedStar Health Research Institute institutional review board (STUDY00002319). The recruitment period started November 23, 2020 and ended March 1, 2021. Participants provided written informed consent [20].

### Study setting

The Curtis National Hand Center in Baltimore, Maryland was selected as the setting for this research for several reasons. It is one of the largest hand centers with the highest volume of hand patients in the nation. The clinic has broad experience with PRO data collection and management. PROs are either self-administered or administered by clinic staff via a web-based system, with over 90% of patients completing data entry and questionnaires [21]. These PROs are used for research and for direct clinical care including patient education and progress tracking. The patient population draws largely from Baltimore City, ranked last for health outcomes among Maryland's 24 counties [22]. Baltimore's prevalence of illiteracy is 15.9% [22] and prevalence of poverty is 22.4% [23], both above the national average.

### Instrument selection

For this first effort to adapt a text-based PROM to its corresponding mPROM, we selected a PROM that measures physical function because questions related to physical status or ability are more conducive to visual representations. We chose the Patient Reported Outcomes Measurement Information System Upper Extremity Short Form 7a (PROMIS-UE) [24], a 7-item questionnaire drawn from the PROMIS-UE Item Bank v2.0. The PROMIS-UE is a generic measure that can be used across all diseases and conditions. It is validated to capture patient perception of upper extremity physical status or ability. It has demonstrated sufficient construct validity with moderate to strong correlation with the legacy instrument Quick Disabilities of Arm, Shoulder, and Hand (QuickDASH) ($r$ = -0.47 to -0.83) [25, 26], good internal consistency (Cronbach alpha coefficient ranging between 0.78–0.93) [27–30], and good responsiveness (AUC 0.81–0.82) [29, 30]. Regarding content validity, PROMIS-UE has not

been found to be limited by floor effects, but there are mixed findings for ceiling effects [29, 31–36].

We chose this measure because it offers several unique advantages. PROMIS instruments, unlike other upper extremity PROMs, are included among the "Common Data Elements" the National Institutes of Health encourages for broad use in research [37]. Additionally, because the PROMIS-UE Item Bank includes questions developed to have the same measurement properties (i.e., measure the same dimension and fit the item response theory model on which they were calibrated), any subset of questions from the bank can be used and scored appropriately [38–40].

## Study design

We used a community-engaged, human-centered design (HCD) approach. Our process for forward adaptation included seven phases, detailed below and summarized in Table 1. The findings and deliverables from each phase were used to inform and refine subsequent phases. This study adheres to our published protocol [13] and is reported according to the Standards for Reporting Qualitative Research (SRQR) [20] as well as Bazzano et al.'s reporting recommendations for health research involving design [41].

**A. Community-engaged study planning.** To obtain community input beyond individuals directly associated with the study, we engaged an eight-member community advisory board (CAB) comprising of racially and ethnically diverse patients and clinical and translational research experts. We presented CAB members with our proposed research plan to obtain feedback with particular attention to optimizing participant recruitment. This feedback and recommended solutions guided the development of the final research plan.

**B. Direct observation.** The lead researcher (CLA), a surgeon trained in qualitative methodologies, conducted direct observation [42] of PROM administration and completion in the clinic's waiting and check-in areas. PROMs were either self-administered or administered by clinic staff, via paper-and-pencil or tablet depending on patient preference. To maximize variation, observations were conducted at different times of the day spanning full clinic operating hours, on different days of the week corresponding with different physicians' clinics, and from

**Table 1. Study design.**

| Phase | Objective | Methods | Data Collected |
|---|---|---|---|
| A. PLANNING | To involve diverse community members and content experts to optimize study processes | Discussed the CAB's concerns with the proposed research plan | Insights that informed research plan development |
| B. OBSERVATION | To understand PROM completion processes and pain points | Direct observation of PROM completion | Insights that informed interview guide development |
| C. DISCOVERY INTERVIEWS | To understand perceptions of and challenges with PROM completion | Semi-structured interviews with patients, caregivers, and clinic staff (purposively sampled low literacy patients) | Personas with user segmentation; themes to guide ideation |
| D. IDEATION | To generate ideas that address design challenges | A workshop utilizing design exercises and "How Might We...?" questions to focus ideation | Key features used as "building blocks" for prototype |
| E. PROTOTYPING | To synthesize features and feedback into mPROMIS-UE prototypes | Worked with illustrator and developer to generate prototypes | Refined prototype |
| F. MEMBER-CHECKING INTERVIEWS | To ensure the completeness and credibility of our findings and obtain early feedback | Semi-structured interviews with patients and caregivers | Resonance of themes and prototype features with participants |
| G. COMMUNITY FEEDBACK | To obtain feedback on prototypes from the community | Discussed acceptability of prototypes with CAB | Feedback used to refine prototype and generate final mPROMIS-UE |

CAB, community advisory board; mPROMIS-UE, multimedia Patient Reported Outcomes Measurement Information System Upper Extremity; PROM, patient-reported outcomes measure

different vantage points. Observations were documented in written memos and moved into a data collection sheet. Data were qualitatively analyzed using design-oriented, rapid thematic analysis [43–45] including iterative reading, insight identification using inductive and abductive thinking, and affinity clustering [46–48]. Resulting clusters and early insights were reviewed by a second team member (LKB) to solidify inductive themes; these themes became the deductive themes used in the subsequent analysis of discovery interviews.

**C. Discovery interviews.** We conducted semi-structured interviews with three participant groups (patients, caregivers, and clinic staff) to understand PROM usability, instrument design, perceived completion barriers, and completion strategies employed. We recruited adult patients with any hand and upper extremity condition including but not limited to those that are traumatic, infectious, congenital, oncologic, and rheumatologic in nature because this is the population for which the PROMIS Upper Extremity is validated in. Patients who identified as English-speaking and who had previously completed the PROMIS-UE were eligible for recruitment. Participants were screened for eligibility through standard clinic intake forms. Those who met inclusion criteria were approached for participation using a purposive sampling strategy [49] developed to ensure heterogeneity of literacy levels in our cohort. Specifically, we used education as a proxy for literacy and only recruited patients who reported an education level of high school or less. Caregivers accompanying patients eligible for study enrollment were also recruited. Finally, clinic staff involved in PROM completion processes were recruited. Reasons for non-participation and characteristics of the non-participant cohort were not recorded. All participants underwent a written informed consent process before interviews were conducted and were compensated $50 USD for their participation.

The goal of each step of HCD research is to identify insights that can be used in design decision-making, rather than to reach thematic saturation as is typically the goal for other qualitative research [50]. Because the rigor of HCD is derived from iterative processes rather than from saturation, we planned to recruit approximately ten participants for this phase [51, 52]. Fewer or more participants were recruited depending on how informative and rich interviews were.

Interviews were conducted by CLA, a researcher with no clinical relationship with patients and no employment relationship with clinic staff, either in-person or over the phone depending on participant preference during the COVID-19 pandemic. We used interview guides tailored to each participant group (S1 Dataset). To assess patient and caregiver participants' literacy level, we administered the Rapid Estimate of Adult Literacy in Medicine Short Form (REALM-SF); a score of 6 or less, corresponding to a sixth grade reading level or less, indicated low literacy [53, 54]. Interviews were audio-recorded and documented using analytic memos written by the interviewer (CLA) within 24 hours after each interview. Subsequent analyses were conducted jointly by two research team members (CLA and LKB) to support reflexivity during analysis. Personas, a HCD technique that aims to summarize each user segment [55], were identified using iterative reading of memos and affinity clustering [47]. We conducted rapid framework-guided thematic analysis [43–45, 56] using the deductive themes identified during observation and identified additional themes inductively via affinity clustering and categorization. These themes were distilled into design challenges that were used to guide ideation. Our rapid thematic analysis aligns with the study's HCD approach. Microsoft Word and Excel software (Redmond, WA) were used to conduct analyses.

**D. Ideation workshop.** Participants in the ideation workshop included five members of our multidisciplinary team including domain experts, clinicians, and designers. Each design challenge identified in the prior interview phase was conceptualized as a "How Might We. . .?" (HMW) question [57]. The workshop included a review of research goals, presentation of findings, individual and group brainstorming around HMW questions, and final discussion. Although patients did not participate in the ideation workshop, we engaged co-creation and

participatory processes [58, 59] by incorporating direct patient quotes and patient suggestions into the discussion. All ideas for potential prototype features were recorded using Miro, a visual collaboration software (San Francisco, California).

**E. Prototyping.** Ideas from the workshop were translated into features that were synthesized into low-fidelity prototypes. Animations were illustrated as storyboards before execution. Prototypes and storyboards were produced in Figma (San Francisco, California). Prototypes underwent three rounds of iterative refinement by the research team, with the goal of improving multimedia components' fidelity to PROMIS-UE's questions and better meeting the design challenges.

**F. Member-checking interviews.** To ensure the completeness, credibility, and validity of our findings, thereby increasing the rigor of our work [60], we conducted additional interviews that included member-checking of findings from prior phases. Only patients and caregivers were recruited, and participant eligibility and recruitment processes were the same as for discovery interviews. The interview guides were updated to include a section presenting and eliciting feedback on findings from prior phases. We aimed to recruit a minimum of five participants, sufficient for one phase of HCD testing [52].

**G. Community feedback.** In our final phase of forward adaptation, we presented the near-final prototype to the CAB, elicited feedback on issues surrounding acceptability, and further refined the prototype to address those issues. This round of refinement incorporating the CAB's feedback was the fourth and final round.

## Results

### A. Community-engaged study planning

The CAB provided several key recommendations that guided research planning. First, the inclusion criteria surrounding English fluency was relaxed to include participants who identify as English-speaking, even if not their primary language or if they were not perfectly fluent. Second, the CAB recommended including caregivers as an additional stakeholder and participant group. Finally, our initial participant compensation of $25 was deemed inadequate, so this was increased to $50.

### B. Direct observation

We conducted 12 hours of observations over a two-week period. We identified eight themes surrounding PROM completion: 1) Survey fatigue: the extent to which it exists, what contributes to it, and how technology amplifies or mitigates it. 2) Patient-centered goal of PROMs: when is this compromised due to caregiver and clinic staff involvement? 3) Timing: how do factors surrounding timing influence how PROMs are completed? 4) Setting: what factors are involved in whether patients complete the survey at home or in the clinic, and which is preferred by patients, caregivers, and staff? 5) Role of the caregiver: in what specific ways do caregivers help or hinder PROM completion? 6) Stakeholder buy-in: in what ways is caregiver, patient, and clinic staff buy-in surrounding PROM completion limited? 7) Role of technology: how do the various technology options facilitate or create barriers to PROM completion, and how can they be leveraged? 8) Help: did patients not ask for help because they did not need help or because they did not want to ask for it?

### C. Discovery interviews

We interviewed 11 participants (7 patients, 2 caregivers, 2 staff) during discovery interviews and 6 participants (5 patients, 1 caregiver) during member-checking interviews. Table 2

**Table 2. Interview participant characteristics.**

| Characteristic | | N (%) or Median (Range) |
|---|---|---|
| TOTAL | | 17 (100) |
| STUDY PHASE | | |
| | Discovery interviews | 11 (65) |
| | Member-checking interviews | 6 (35) |
| PARTICIPANT CATEGORY | | |
| | Patient | 12 (71) |
| | Caregiver | 3 (18) |
| | Clinic staff | 2 (12) |
| SEX | | |
| | Female | 10 (59) |
| | Male | 7 (41) |
| AGE (YEARS) | | 44.5 (20–81) |
| INTERVIEW SETTING | | |
| | In-person | 5 (29) |
| | Telephone | 12 (71) |
| LITERACY (REALM-SF) SCORE | | 7 (0–8) |
| LITERACY CATEGORY | | |
| | Low | 6 (35) |
| | High | 7 (41) |
| | Unknown | 4 (24) |

REALM-SF, Rapid Estimate of Adult Literacy in Medicine Short Form

provides the demographics and characteristics for our entire cohort of 17 participants. Importantly, our cohort was of mixed literacy, with approximately half (46%) scoring low literacy on the REALM-SF.

**Personas.** Our interviews revealed two distinct user personas: 1) the 'get-it-done' persona, i.e., the patient who wants to complete the PROMs as quickly as possible and just wants to get it over with; and 2) the 'do-it-right' persona, i.e., the patient who is thoughtful, wants to make sure he/she completes it accurately, may need more time to complete it, and may want help understanding the questions correctly. Our interviews also revealed three literacy personas: 1) the 'illiterate' persona, i.e., low literacy to the extent that they cannot complete the form independently; 2) the 'slow reader' persona, i.e., not completely illiterate but sufficiently low literacy that they struggle with PROMs, spending a long time in the waiting room attempting to complete it; and 3) the 'literate' persona, i.e., literacy level does not pose a challenge to PROM completion. We aimed to design a mPROMIS-UE that is well-suited for all user and literacy personas.

**Location.** Between completing the PROMs at home versus at the clinic, the overwhelming preference by all participant groups was completion at home. The reasons cited included comfort, convenience, ability to take as much time as needed, increased privacy, minimizing time needed in a hospital/clinic, and avoidance of disrupting clinic flow. There were still instances in which patients were asked to complete their PROMs in clinic if they did not complete them before their appointment. We concluded that the mPROMIS-UE should allow for completion in both a home and a clinic setting.

**Length and timing.** All participant groups, across user and literacy personas, highlighted the importance of a mPROMIS-UE that does not increase the time needed for completion.

**Platforms and technology.** All participant groups preferred digital PROMs over paper PROMs. Smartphones, tablets, computers, or laptops were utilized preferentially by different patients and for different reasons, suggesting that the mPROMIS-UE should be adaptable to various digital platforms.

**Customizability.** There was a wide range of reactions to the possibility of including multimedia audiovisual components. Some participants thought these components would be helpful, specifically for low literacy or cognitively impaired individuals. Other participants thought that PROMs were straightforward and would not benefit from multimedia components. Several worried that the addition of multimedia components, especially videos, would make the survey longer to complete. This highlighted the importance of designing an instrument that fits the needs of all user and literacy personas. Additionally, staff participants highlighted that they disliked having to administer more than one mode of PROM completion (e.g., paper and pencil vs. tablet vs. staff-administration), and preferred one uniform, streamlined process. Creating a single mPROMIS-UE that is customizable based on patient need or preference would address both patient and staff concerns.

**Anxiety and shame.** Both anxiety and shame were themes associated with PROM completion. The first was anxiety due to unfamiliarity with PROMs. One patient explained, "When I first got the tablet in my hand, a little anxiety came over me because I'm not much into computers." The second was shame or discomfort surrounding difficulty completing PROMs. One staff participant explained, "For the ones that don't really read, you have some in their 50s or 60s that don't really mind telling you that, but those ones in their 70s or 80s with their pride won't tell you that, they'll just say 'I can't do that.'" These findings reinforced the importance of creating a customizable mPROMIS-UE for all literacy levels that avoids the need to identify low literacy patients and minimizes anxiety, shame, and discomfort.

**Caregivers' roles in PROM completion.** We found that there are several types of caregivers, including family members, friends, nurses or nurse's aides, transportation staff, and house managers. Caregivers assisted with PROM completion in various capacities, such as with holding the device, reading questions and responses aloud, clarifying the meaning of questions, filling in answers, or troubleshooting navigation-related issues. Caregivers' ability to assist with PROM completion was variable, sometimes influenced by their own literacy or ability. This suggested that the mPROMIS-UE could be helpful even for patients who have caregivers.

## D. Ideation workshop

The themes and insights from interviews were distilled into four design challenges: literacy, customizability, convenience, and avoiding discomfort (Table 3). Each idea for addressing the

**Table 3. Design challenges and corresponding "How Might We...?" questions and ideas.**

| Design Challenge | "How Might We...?" Question | Idea Names |
|---|---|---|
| Literacy | How might we design a PROM that can be self-administered by all literacy personas? | "I-con believe it," "Just say it," "Multiple channels," "QOL-oji" |
| Customizability | How might we design a PROM that can discern and be responsive to differing user and literacy personas? | "Smart PROM," "B(udget friendly)-PROM," "On/off toggle", "Avatars" |
| Convenience | How might we design a PROM that is convenient (from timing, location, platform perspectives)? | "Smooth-e/seamless," "Easy platform for data transfer," "Manipulable" |
| Avoiding Discomfort | How might we design a PROM that avoids any potential to cause patients/caregivers anxiety or shame? | "Everywhere you click," "Select survey version," "Take it at home," "Avatar" |

PROM, patient-reported outcomes measure

design challenge was given a specific name to facilitate identification and discussion of the various ideas (final column, Table 3).

## E. Prototyping

Through three rounds of iterative review and discussion among the research team, ideas from the workshop were translated into six features that were built into a low-fidelity prototype: avatars, audio, GIFs (i.e., animated cartoon), an illustrated response scale (specifically, the COOP/ WONCA overall health response scale that has been widely validated in low literacy populations [61]), icons to deploy audio and GIFs, and a progress indicator. The prototype was developed for a smartphone platform because this best addressed the location aspect of the convenience design challenge. Table 4 details which design challenge(s) each prototype feature was designed to address, how it did so, and examples of how these decisions were informed by findings from prior phases.

## F. Member-checking interviews

The last six interviews included member-checking. Participants felt that the themes and design challenges resonated with them and were complete; this validated and confirmed the

**Table 4. Prototype features.**

| Design Challenge | Prototype Feature That Addresses the Challenge | How Decisions were Informed by Findings from Prior Phases |
|---|---|---|
| Literacy | • Audio: Each question and response has associated audio that reads the text.<br>• GIFs: Each question has an associated animated cartoon that "acts out" the task asked about in the question.<br>• Illustrated response scale: Each response option has a corresponding illustration. | • The availability of audio, GIFs, and the illustrated response scale addresses the needs and preferences of the 'illiterate' and 'slow reader' literacy personas that our discovery interviews revealed.<br>• Their availability is important even for patients who have caregivers, which was also identified during discovery interviews. |
| Customizability | • Icons: The default version that is displayed is the text-based PROMIS-UE. The audio, GIFs, and illustrated response scale are deployed via small icons available for each question. This allows patients the flexibility to choose which multimedia component to activate (if any), and for which questions. | • The ability to customize the experience allows high literacy users who do not need the multimedia features to avoid them, potentially shortening the time of survey completion. This was guided by the 'survey fatigue' and 'timing' themes identified during direct observation and 'length and timing' and 'customizability' themes from discovery interviews. This also caters to the needs and preferences of all the distinct user and literacy personas that our discovery interviews revealed. |
| Convenience | • Progress indicator: This indicates how many questions are remaining so that patients can plan their time accordingly.<br>• Icons: The customizability that the icons afford allow the 'literate' persona to avoid multimedia components, which may increase the time needed to complete the mPROMIS-UE.<br>• Development on smartphone: This prototype was first developed for smartphone use because it is the device that most patients have ready access to. Smartphones' portability also allows patients to complete the mPROMIS-UE at either home or the clinic. | • Convenience afforded by the smartphone platform was guided by the 'setting' and 'role of technology' themes from direct observation, and the 'location' and 'platforms and technology' themes from discovery interviews.<br>• Convenience afforded by the progress indicator provides the 'do-it-right' persona the information to decide when and where is best to complete the mPROMIS-UE. The importance of this was guided by the 'setting' theme from direct observations. |
| Avoiding Discomfort | • Avatar: This includes diverse avatars, among which users can select one. The avatar selected is then the one that appears in the GIFs.<br>• Icons: Using multimedia components, especially in a public setting, may cause discomfort. The customizability afforded by the icons allows patients to decide whether to deploy multimedia components or not. | • Inclusion of avatars was guided by the 'patient-centered goals of PROMs' theme identified in direct observation and 'anxiety and shame' theme that emerged from discovery interviews.<br>• Customizability gives patients the ability to consider how their setting influences their preferences for using or not using multimedia features to avoid discomfort, which we prioritized due to the 'setting' theme from direct observations and 'anxiety and shame' theme from discovery interviews. |

mPROMIS-UE, multimedia Patient Reported Outcomes Measurement Information System Upper Extremity; PROMIS-UE, Patient Reported Outcomes Measurement Information System Upper Extremity

authenticity and completeness of our findings from prior phases. However, the response to potential prototype features varied. For example, while some participants responded enthusiastically to the idea of GIFs, others disliked this idea for various reasons. These responses and preferences reinforced the importance that mPROMIS-UE meets the customizability design challenge.

## G. Community feedback

The CAB suggested that we modify certain features to improve instrument design and the user interface (e.g., make the progress indicator more visual), adjust GIFs so that they were more intuitive or easy to understand, and increase avatar diversity.

Screen captures of the final prototype are shown in Fig 1. This prototype corresponds with "mPROM v1.0" in our published protocol [13]. It provides instructions using text, icons, as well as audio. It asks patients to select among four avatars that are gender neutral but vary in skin tone, in order to represent racial and ethnic diversity. Each question is accompanied by audio and a GIF, and the GIF features the avatar that the patient selected. Each answer response is paired with an illustration, and also has an audio option. The audio, GIFs, and illustrated scale are deployed only if the patient clicks the corresponding icon. A progress bar indicates how many questions have been completed. Unanswered questions are highlighted if the user tries to submit an incomplete form.

## Discussion

By executing forward adaptation, the first stage of the MAP, we successfully adapted the PROMIS-UE to the mPROMIS-UE by addressing the challenges identified by a mixed literacy hand and upper extremity patient cohort. The mPROMIS-UE is customizable and responsive to our five user and literacy personas, convenient in terms of timing, location, and platform, and is designed to avoid potential shame/discomfort. When we shared our initial findings and prototype ideas with patients, caregivers, and our CAB, their responses indicated that our research goals and findings broadly resonated with them. They offered recommendations for improving the mPROMIS-UE in various domains that were integrated into the final mPROMIS-UE prototype. This work demonstrates the feasibility of adapting PROMs to multimedia versions that can be self-administered by mixed literacy populations.

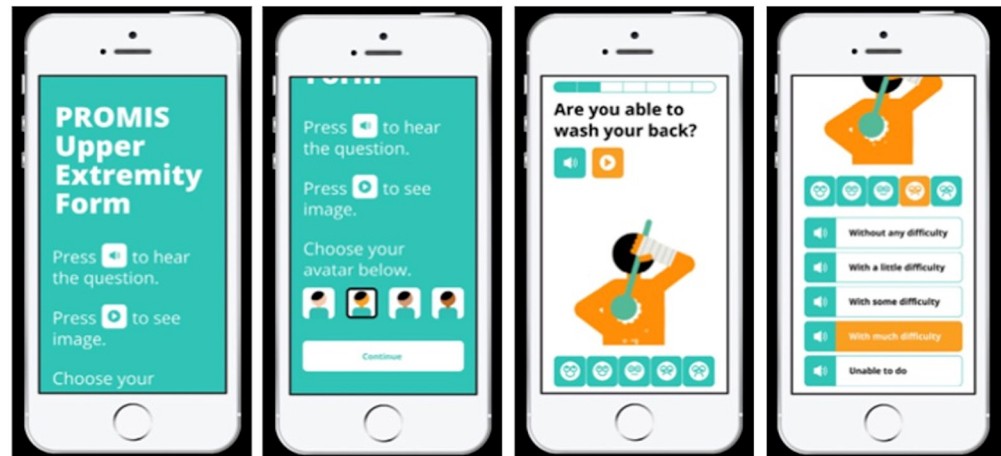

**Fig 1. Multimedia PROMIS upper extremity.** Screen Captures of the multimedia Patient Reported Outcomes Measurement Information System Upper Extremity (mPROMIS-UE).

There is limited literature on developing PROMs for mixed literacy populations. The only attempt to address the challenge of PROMs for mixed literacy populations that has been studied and reported on is the "Talking Touchscreen" (TT). TT is a platform for administering PROMs with audio recordings (but no visuals) [62, 63]. This was initially developed for health-related quality of life assessment in Latino cancer patients with low literacy levels [63]. The PROMIS Global Health is currently the only PROM available for administration via TT [64].

The mPROMIS-UE adds to the limited repository of PROMs (i.e., TT) designed for mixed literacy populations. It however is a prototype and will not be ready for implementation until the remaining stages of the MAP (back adaptation, qualitative evaluation, psychometric validation) [13] have been executed. Back adaptation will ensure that we have not altered the content of the PROMIS-UE during forward adaptation. Qualitative evaluation will ensure a well-designed user interface that increases the likelihood of successful implementation and long-term adoption and minimizes the risk of human error and operating inefficiencies. Psychometric validation is needed to confirm whether the mPROMIS-UE measures the construct it is intended to measure, in the intended population. Once these stages are completed, to advance a multimedia approach to capturing PRO data, we recommend future research efforts focus on dissemination, implementation, and expansion. This can include but is not limited to making mPROMIS-UE an official PROMIS instrument that is publicly available for widespread use, deploying it in hand and upper extremity clinics, integrating with the electronic health record, and collaborating with investigators from other specialties to adapt additional PROMs to mPROMs.

## Implications

There are several implications to this work. To our best knowledge the mPROMIS-UE is the first PROM that incorporates visual components and was designed through a community-engaged and HCD approach. This allows the mPROMIS-UE to be self-administered by mixed literacy patients, who provided input in its development. mPROMIS-UE and the MAP provide a novel option for collecting PROs in English-speaking patients of mixed literacy, in the United States and other English-speaking countries in which the PROMIS-UE is validated. The mPROMIS-UE was designed to be customizable for all distinct user and literacy personas that we identified, which streamlines future implementation. The mPROMIS-UE was also designed to be self-administered by all user and literacy personas. This reduces the need for staff-administration that is labor intensive and expensive [8], and also perceived as undesirable by our patients, caregivers, and clinic staff. As PROs are adopted increasingly in practice and research, mPROMIS-UE and mPROMs broadly may be a preferred, practical, and cost-effective option for all stakeholders. In addition, this study demonstrated the feasibility of the first stage of the MAP, forward adaptation [13].

Although the primary goal of the mPROMIS-UE was to meet the needs of mixed literacy populations, particularly in support of low literacy patients, our findings suggest that there may be secondary benefits. For example, there is known variability in the comprehension and interpretation of the PROMIS-UE questions [65]. Participants noted that multimedia components can help address ambiguity of certain questions, so multimedia components have the potential to improve the validity and reliability of PROMs for use in for the general population. The audiovisual cues in a well-designed mPROM may also facilitate understanding for patients with cognitive impairments. Additional research is needed to explore these potential secondary benefits.

## Limitations

One limitation related to the scope of our study is that we may have missed some relevant perspectives. For example, patients with cognitive disabilities or with poor English proficiency

may have perspectives that differ from the broader low print literacy population. However, because cognitive disabilities and poor English proficiency are related but distinct barriers to PROM completion from low print literacy [66], we chose to focus our recruitment strategy on ensuring variation in print literacy levels. Future research should examine how text-based PROMs pose challenges to cognitively impaired or non-English speaking patients, as well as whether multimedia instruments are an appropriate or ideal solution. Additionally, our community advisory board included only eight members, so our community input beyond those directly associated with the study came from a relatively small number of individuals.

## Conclusions

Development of the mPROMIS-UE is the first step towards expanding clinicians' and researchers' ability to capture PROs in mixed literacy hand and upper extremity patient populations. More broadly, it demonstrates the feasibility of adapting existing text-based PROMs to mPROMs. As such, this study represents an advancement towards a more equitable and inclusive paradigm in PRO measurement whereby the standard instruments are available in user-accessible multimedia versions.

## Supporting information

**S1 Dataset. Minimal data set including observation data, discovery interview guides, ideation workshop outline, and member checking interview guide.**
(DOCX)

## Acknowledgments

We thank Claudia Udler and Hayelin Choi for their invaluable contributions as the social design and graphic design consultants, respectively, of our research team. Our previous protocol was published in PLOS ONE as a Registered Report Protocol entitled, "Developing a protocol for adapting multimedia patient-reported outcomes measures for low literacy patients," and this work is the corresponding Registered Report. The only change to the author list is the addition of Allan Fong in this work.

## Author Contributions

**Conceptualization:** Chao Long Azad, Albert W. Wu, Aviram M. Giladi.

**Data curation:** Chao Long Azad, Aviram M. Giladi.

**Formal analysis:** Chao Long Azad, Laura K. Beres, Aviram M. Giladi.

**Funding acquisition:** Chao Long Azad, Aviram M. Giladi.

**Investigation:** Chao Long Azad, Aviram M. Giladi.

**Methodology:** Chao Long Azad, Laura K. Beres, Albert W. Wu, Allan Fong, Aviram M. Giladi.

**Project administration:** Chao Long Azad, Aviram M. Giladi.

**Resources:** Chao Long Azad, Aviram M. Giladi.

**Software:** Allan Fong.

**Supervision:** Chao Long Azad, Aviram M. Giladi.

**Validation:** Chao Long Azad, Laura K. Beres, Albert W. Wu, Allan Fong, Aviram M. Giladi.

**Visualization:** Chao Long Azad, Allan Fong, Aviram M. Giladi.

**Writing – original draft:** Chao Long Azad, Aviram M. Giladi.

**Writing – review & editing:** Chao Long Azad, Laura K. Beres, Albert W. Wu, Allan Fong, Aviram M. Giladi.

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
