## [Decision Letter · Decision Letter 0]

21 Nov 2023

PONE-D-23-24670Developing a multimedia patient-reported outcomes measure for low literacy patientsPLOS ONE

Dear Dr. Giladi,

Thank you for submitting your manuscript to PLOS ONE. After careful consideration, we feel that it has merit but does not fully meet PLOS ONE’s publication criteria as it currently stands. Therefore, we invite you to submit a revised version of the manuscript that addresses the points raised by the two reviewers during the review process.

We look forward to receiving your revised manuscript.

Kind regards,

Xiaodan Tang

Academic Editor

PLOS ONE

Journal Requirements:

3. We note that Figure 1 in your submission contain copyrighted images. All PLOS content is published under the Creative Commons Attribution License (CC BY 4.0), which means that the manuscript, images, and Supporting Information files will be freely available online, and any third party is permitted to access, download, copy, distribute, and use these materials in any way, even commercially, with proper attribution. For more information, see our copyright guidelines: http://journals.plos.org/plosone/s/licenses-and-copyright.

Reviewers' comments:

Reviewer's Responses to Questions

**Comments to the Author**

1. Does the manuscript adhere to the experimental procedures and analyses described in the Registered Report Protocol?

If the manuscript reports any deviations from the planned experimental procedures and analyses, those must be reasonable and adequately justified.

Reviewer #1: Yes

Reviewer #2: Yes

2. If the manuscript reports exploratory analyses or experimental procedures not outlined in the original Registered Report Protocol, are these reasonable, justified and methodologically sound?

A Registered Report may include valid exploratory analyses not previously outlined in the Registered Report Protocol, as long as they are described as such.

Reviewer #1: Yes

Reviewer #2: Yes

3. Are the conclusions supported by the data and do they address the research question presented in the Registered Report Protocol?

The manuscript must describe a technically sound piece of scientific research with data that supports the conclusions. The conclusions must be drawn appropriately based on the research question(s) outlined in the Registered Report Protocol and on the data presented.

Reviewer #1: Yes

Reviewer #2: Partly

4. Have the authors made all data underlying the findings in their manuscript fully available?

Reviewer #1: Yes

Reviewer #2: Yes

5. Is the manuscript presented in an intelligible fashion and written in standard English?

Reviewer #1: Yes

Reviewer #2: Yes

6. Review Comments to the Author

Please use the space provided to explain your answers to the questions above. (Please upload your review as an attachment if it exceeds 20,000 characters)

Reviewer #1: This report describes the development of an English language multimedia version of the PROMIS-UE. The project implemented rigorous methodology, consisting of seven phases, and feedback from a community advisory board, patients, caregivers, and clinic staff. Future research will include back adaptation, usability testing via qualitative evaluation, and psychometric validation of the mPROMIS-UE. If successful, this multimedia instrument would be a useful addition to collecting patient-reported outcomes in diverse literacy populations.

There are only a few issues that need some additional details:

1. There is no description of the patient population, other than to state that they had “a hand and upper extremity condition.” It is not clear what this means. Also, how large is this population in the U.S.?

2. What mode and method of administration were used for the direct observation phase of PROM administration and completion in the clinic’s waiting and check-in areas?

3. The author chose the Patient Reported Outcomes Measurement Information System Upper Extremity Short Form 7a (PROMIS-UE). A definition of this measure is needed.

Reviewer #2: This well-written paper essentially describes a human-centered design process. On the whole, it is overly focused on methods (HCD) and not focused enough on results. In the abstract, the first five sentences of "results" belong with the methods. The development of the mPROMIS-UE prototype is the result - not the "identification" of personas. ("Identification" seems an overly strong word choice. My impression is that your observations and interviews informed the profiles and personas that you used to develop the modified tool.)

I have a parallel comment for the body of the paper. I recommend reflecting on what is "methods" and what is "results," and how to shift to a more appropriate balance. I imagine that you're hoping to convey to readers the fruit of your effort, rather than the process of developing it.

In your introduction, you write that your first goal was to create the mPROMIS-UE, and the second goal was to assess its acceptability. Were those the goals of this paper, or the project overall? If the former, check to ensure that the paper's focus is on these objectives.

Your community advisory board does not have very many members. It may be helpful to contextualize its size with reference to work by others, and to consider listing as a limitation that your user feedback came from a relatively small number of users.

What is the relevance for low-literacy populations outside of the US? Could your tool be employed elsewhere, or just with the population for which it was designed through HCD?

7. PLOS authors have the option to publish the peer review history of their article (what does this mean?). If published, this will include your full peer review and any attached files.

Reviewer #1: No

Reviewer #2: No

---

## [Author Response · Author response to Decision Letter 0]

8 Jan 2024

We have included rebuttals to each reviewer comment in the attached table of corrections/response to reviewers. Please see the attached document.

---

## [Decision Letter · Decision Letter 1]

26 Feb 2024

PONE-D-23-24670R1Developing a multimedia patient-reported outcomes measure for low literacy patientsPLOS ONE

Dear Dr. Giladi,

Thank you for submitting your manuscript to PLOS ONE. After careful consideration, we feel that it has merit but does not fully meet PLOS ONE’s publication criteria as it currently stands. Therefore, we invite you to submit a revised version of the manuscript that addresses the points raised during the review process.

We look forward to receiving your revised manuscript.

Kind regards,

Xiaodan Tang

Academic Editor

PLOS ONE

Journal Requirements:

Reviewers' comments:

Reviewer's Responses to Questions

**Comments to the Author**

1. Does the manuscript adhere to the experimental procedures and analyses described in the Registered Report Protocol?

If the manuscript reports any deviations from the planned experimental procedures and analyses, those must be reasonable and adequately justified.

Reviewer #2: Yes

Reviewer #3: Partly

2. If the manuscript reports exploratory analyses or experimental procedures not outlined in the original Registered Report Protocol, are these reasonable, justified and methodologically sound?

A Registered Report may include valid exploratory analyses not previously outlined in the Registered Report Protocol, as long as they are described as such.

Reviewer #2: Yes

Reviewer #3: Partly

3. Are the conclusions supported by the data and do they address the research question presented in the Registered Report Protocol?

The manuscript must describe a technically sound piece of scientific research with data that supports the conclusions. The conclusions must be drawn appropriately based on the research question(s) outlined in the Registered Report Protocol and on the data presented.

Reviewer #2: Yes

Reviewer #3: Yes

4. Have the authors made all data underlying the findings in their manuscript fully available?

Reviewer #2: Yes

Reviewer #3: Yes

5. Is the manuscript presented in an intelligible fashion and written in standard English?

Reviewer #2: Yes

Reviewer #3: Yes

6. Review Comments to the Author

Please use the space provided to explain your answers to the questions above. (Please upload your review as an attachment if it exceeds 20,000 characters)

Reviewer #2: Thank you for your edits to strengthen the manuscript. It is much better than your last version. Thank you for your responses to my observations and recommendations.

I appreciate that you changed some instances of persona development to “…our interviews revealed two distinct user personas…” I would call attention to the reality that your abstract still retains the use of “identify,” asserting that you “identified two distinct user personas…” As in my last review, I find this word choice to be overly strong, since another HCD process could conceivably reveal alternative personas. In keeping with the qualitative nature of this undertaking, I wonder if you would please locate continued uses of “identify” in relation to personas and change to a different verb.

I have no additional comments.

Reviewer #3: Thank you for the opportunity to review this manuscript, which describes and important development in the field of patient-reported outcomes measurement. I hope the following comments will help to further improve the manuscript. I organized the comments by section and, where applicable, line numbers.

ABSTRACT

28: The authors may wish to instead refer to their previously published Multimedia Adaptation Protocol.

53-54: In the abstract, the authors refer to “feasibility of adapting PROMs to multimedia versions”. As I read the manuscript, I understand this to be the primary goal. Please consider clarifying the goal in the manuscript.

57: Please delete the term “reliably”. The results do not provide any evidence of reliability and reliability is not discussed in the manuscript.

INTRODUCTION

60-61: Given the broad readership, it is important to explain what PROs are. Please provide a definition so readers can understand what kind of outcomes this refers to.

76: “to accurately capture PROs”. Is accuracy the main goal? The study seems to focus more on accessibility, appropriateness, and feasiblity, which are important goals from an equity point of view. Please consider clarifying this here.

77: The specific focus on “hand and upper extremity patient population” comes as a surprise, given the flow of the argument in the discussion. Please explain in the introduction why you are focusing on this particular measure and patient population. If the intent is for this to serve as a context for demonstrating the application of the protocol, please say so. Or, else, provide another rationale for this specific focus.

Additionally, given the broad readership of this journal, please describe further in the introduction what is meant by "hand and upper extremity patients" and provide some description of how PROMs are being used in this context (e.g., To determine the best treatment options? Or to evaluate the outcomes of surgical or medical interventions?).

81: “MAP is an innovative protocol for adaptation”. This reads rather repetitive, stating the obvious. Please consider revising.

METHODS

Overall comment: Please refer to the journal's guidelines on reporting of qualitative research. A few key points are identified in my comments above. https://journals.plos.org/plosone/s/submission-guidelines#loc-guidelines-for-specific-study-types: "Qualitative research studies should be reported in accordance to the Consolidated criteria for reporting qualitative research (COREQ) checklist or Standards for reporting qualitative research (SRQR) checklist."

Study setting (101): To address my comment above, please explain how PRO data are currently being used in this clinic. What are the purposes for collecting the PRO data?

Instrument selection (113): Please provide further description of the items and measurement validity evidence. Given the focus of this manuscript, it is important to fully understand what is being measured and what the validity evidence of the original instrument is.

Study design

141: Please provide a reference to participant observation methods.

149: Please define what this means. How is this different from regular thematic analysis? The description of the results reflect a more general content analysis approach. Please describe how the data were analyzed to arrive at themes. Or, if content analysis was used, please indicate this. Either way, please provide a suitable reference for the approach that was used.

Discovery interviews: Some inclusion criteria are described, but the sampling approach is unclear. Please describe how participants were selected and the type of sampling method used (e.g., purposive, convenience, consecutive, snowball). Did any of the approached people refuse to participate? If so, is anything known about the reasons for non-participation?

171: Please indicate here that the CLA was the interviewer and describe the nature of the relationship of CLA with the interview participants. Is it possible that he results may have been influenced by the relationship.

172: Please provide further description of the interview guides (e.g., provide examples of questions asked or include the guides as an appendix).

178: Was CLA the only person doing the analyses, or were others involved in the analysis as well? Please describe.

179: Please provide a reference to the thematic analysis approach that was used (given that there are different methodological approaches - see above comment). Please also indicate whether any software was used to assist with the analysis.

Ideation workshop: It seems that patients were not included in this phase. Is that correct? Please make this explicit, as this could be viewed as a limitation.

RESULTS

Direct observation: See comment above regarding content vs. thematic analysis. The results, as presented here, seem to be more representative of a type of content analysis. It is unclear what the inductive themes might be. Please revise or clarify.

Ideation workshop: It is unclear what all the idea names in Table 3 refer to. Please provide further description in the text or a footnote to this table.

Prototyping: Based on the MAP approach, I understand that the prototype should be informed by the results of the prior phases (B, C and D). However, this is currently not clear. Please expand the text, or in the table, to more explicitly demonstrate how the prototype was developed based on the results reported above.

DISCUSSION

It would be helpful, and conventional, to first describe a summary of the overall result of your study.

365: Please discuss what you consider to be the key recommendations for advancing this approach to PRO data capture.

It is critically important to indicate the need for further research on validity evidence. The authors correctly refer to this in the abstract, when they state: "“Future research will include back adaptation, usability testing via qualitative evaluation, and psychometric validation of the mPROMIS-UE”. Please elaborate on this in the discussion section of the manuscript.

7. PLOS authors have the option to publish the peer review history of their article (what does this mean?). If published, this will include your full peer review and any attached files.

Reviewer #2: No

Reviewer #3: No

---

## [Author Response · Author response to Decision Letter 1]

10 Apr 2024

All comments have been addressed in the attached Response to Reviewers table.

---

## [Editor Report · Decision Letter 2]

15 Apr 2024

PONE-D-23-24670R2Developing a multimedia patient-reported outcomes measure for low literacy patients with a human-centered design approachPLOS ONE

Dear Dr. Giladi,

Thank you for submitting your manuscript to PLOS ONE. After careful consideration, I think there are still some minor revisions (see below) needed before it can be accepted for publication.  1. The first mention of PROMIS needs to include a trademark, PROMIS®

2. PROMIS UE is not disease-specific. It is a generic measure that can be used across all diseases and conditions. Please revise accordingly in the "Instrument selection" section.3. In the "Implications" section, the authors stated that "as the mPROMIS-UE could replace the current PROMIS instrument rather than being an additional instrument to be administered to certain segments of the clinic population." This statement was not supported by the evidence provided in this paper, as no thorough psychometric analysis has been conducted for mPROMIS-UE. 

We look forward to receiving your revised manuscript.

Kind regards,

Xiaodan Tang

Academic Editor

PLOS ONE
---

## [Author Response · Author response to Decision Letter 2]

19 Apr 2024

Please see response to reviewers doc in the submission.

---

## [Editor Report · Decision Letter 3]

10 May 2024

Developing a multimedia patient-reported outcomes measure for low literacy patients with a human-centered design approach

PONE-D-23-24670R3

Dear Dr. Giladi,

We’re pleased to inform you that your manuscript has been judged scientifically suitable for publication and will be formally accepted for publication once it meets all outstanding technical requirements.

Kind regards,

Xiaodan Tang

Academic Editor

PLOS ONE
---

## [Editor Report · Acceptance letter]

27 May 2024

PONE-D-23-24670R3 

PLOS ONE

Dear Dr. Giladi, 

I'm pleased to inform you that your manuscript has been deemed suitable for publication in PLOS ONE. Congratulations! Your manuscript is now being handed over to our production team.

Kind regards, 

on behalf of

Dr. Xiaodan Tang 

Academic Editor

PLOS ONE